# Characteristics and Drivers of Soil Organic Carbon Saturation Deficit in Karst Forests of China

Limin Zhang [1,2], Yang Wang [1], Jin Chen [1], Ling Feng [1], Fangbing Li [1] and Lifei Yu [1,*]

1   College of Life Sciences, Guizhou University, Guiyang 550025, China; zhanglimin563406@163.com (L.Z.); wangyang20218910@126.com (Y.W.); gs.chenjin20@gzu.edu.cn (J.C.); 15870149772@163.com (L.F.); lifangbing9685@163.com (F.L.)
2   Institute of Guizhou Mountain Resources, Guizhou Academy of Sciences, Guiyang 550001, China
*   Correspondence: lfyu@gzu.edu.cn

**Abstract:** Karst forests have complex and unique carbon cycle characteristics. Soil organic carbon saturation deficit (CSD) is an important indicator of soil organic carbon (SOC) sequestration potential; exploring its characteristics and driving factors is a priority theme in current research on the carbon cycles of terrestrial ecosystems. In this study, 171 topsoil samples from typical karst forests in southwest China were used as the study objects. A SOC maximum saturation capacity model was constructed using the boundary line method. The CSD is equal to the maximum saturated capacity of SOC minus the current SOC. We analyzed the CSD and its main driving factors in different regions and succession stages. The results showed that the fractions of carbon and SOC contents in the karst forests at different successional stages in descending order were as follows: climax stage > arbor stage > shrub stage > herb stage. The CSD was the highest at the herb stage in Maolan, Yuntai Mountain, and Dashahe at 83.04%, 89.99%, and 89.97%, respectively, followed by the shrub stage with 48.69%, 78.50%, and 84.95%, and the lowest at the arbor stage with 25.69%, 43.44%, and 60.49%. The main drivers of CSD in the karst forest of Maolan were litter carbon input, total nitrogen, total phosphorus, and total SOC, and were litter carbon input at Yuntai Mountain and litter carbon input and neutral phosphatase at Dashahe. The results indicate that the core driver of CSD in the karst forest is litter carbon input, and this can be adjusted in the future to regulate the carbon sequestration capacity of SOC.

**Keywords:** karst forests; soil organic carbon; driving factors; carbon sequestration potential

## 1. Introduction

Soils are considered the largest terrestrial pool of global carbon. Soil organic carbon (SOC) pool is an important and variable carbon reservoir in terrestrial ecosystems and a potential sink of greenhouse gases that can exhibit significant spatial variation [1,2]. The forest SOC pool is an important component of the forest ecosystem; 73% of global soil carbon is stored in forest soils. Therefore, it plays an irreplaceable role in maintaining the global climate system, regulating the global carbon balance, and slowing the rise of atmospheric greenhouse gas concentrations [3,4]. Small changes in this carbon pool, especially in the mineral particle organic carbon (<53 μm), will affect the global carbon balance and lead to global climate change [5].

Numerous studies have found that SOC eventually reaches equilibrium as exogenous carbon inputs continue to increase [6–8], i.e., soil carbon saturation [9]. Stewart et al. [10] and Feng et al. [11] estimated the maximum saturation of SOC in grassland, agricultural land, and forest by constructing a model of the maximum saturation capacity of SOC, thereby providing a foundation for subsequent studies on the carbon sequestration potential of SOC and the relevant influencing factors. In recent years, research on SOC in forest ecosystems has focused on the sequestration of SOC, decomposition of litter, and the influence of environmental factors on SOC at large spatial scales [12,13]. Zhou et al. [14]

found that soil carbon content in China's forest ecosystems accounted for 3/4 of the total forest ecosystem; Huang et al. [15] found that SOC content increased with the gradual recovery of vegetation, and soil carbon sequestration capacity was enhanced through the study of different vegetation restoration processes in Maolan. Zhang et al. [16] and Lal et al. [17] found that changes in soil carbon flux were mainly influenced by the interaction of vegetation, climate, and soil properties. Soil organic carbon saturation deficit (CSD) is equal to the maximum saturated capacity of SOC minus the current SOC [10], which is an important index directly reflecting soil carbon sequestration potential. Di et al. [18] found that in agricultural soils, increased application of organic fertilizers significantly reduced CSD over time, resulting in less space for future carbon sequestration, and carbon input was the main influence factor on CSD. At present, there have been few studies on the characteristics and drivers of saturation deficit in forest soils.

Karst forests are forest ecosystems that are distributed on landscapes with limestone, dolomite, and carbonate as the main bedrock. Plants in karst forests are constrained by soil topography [19,20]; as a unique ecosystem, the topography, hydrothermal conditions, and soil development conditions differ from those of non-karst areas [21–24]. Based on this complexity and specificity, the study of CSD in karst forest ecosystems is highly relevant. The key questions addressed in this study were: (1) What are the levels of CSD at different successional stages in karst forests, and how does this affect the future carbon sequestration potential? (2) What are the main drivers of CSD in karst forests, and how can they be regulated?

## 2. Materials and Methods

### 2.1. Overview of the Study Areas

The study areas were located in Maolan National Nature Reserve, Shibing Yuntai Mountain Nature Reserve, Daozhen Dashahe Nature Reserve, Nayong Gongtong Nature Reserve, Pogang Karst Vegetation Nature Reserve, Kuankuoshui Nature Reserve, Puding Huoyan Mountain Nature Reserve, Jiangkou Huanggu Mountain Nature Reserve, Kaiyang Zijiang Geosuture, and Wangmo Bijia Mountain, all in Guizhou Province, China (Figure 1). These study areas are typical karst forests that present climax communities because they are long-established forests that experience low levels of disturbance, and thus they can be used to model the maximum saturation capacity of SOC in karst forests. Among these karst forests, three areas with rich successional stages were selected from south to north, namely, Maolan National Nature Reserve, Shibing Yuntai Mountain Nature Reserve, and Daozhen Dashahe Nature Reserve, to explore CSD and the driving factors at different successional stages. The Maolan National Nature Reserve has a total area of 213 km$^2$ with a maximum elevation of 1079 m, a minimum elevation of 430 m, and an average elevation of 700 m. It has a central subtropical southern monsoon climate with an average annual temperature of 18.3 °C, annual precipitation of 1321 mm, and an annual sunshine duration of 1271 h. Most parts of the reserve are central subtropical primary karst forests, with mixed evergreen, deciduous broad-leaved tree species. There are different degrees of successional communities, with 1203 species of vascular plants in 154 families and 514 genera [15]. The Yuntai Mountain Nature Reserve has a total area of 47 km$^2$ with a maximum elevation of 1869 m, a minimum elevation of 486 m, and an average elevation of 526 m. It has a humid subtropical monsoon climate with an average annual temperature of 16.4 °C, annual precipitation of 1130 mm, and an annual sunshine duration of 1197 h. The reserve has the typical karst topography of southern China; the area is also a World Heritage Site with vegetation growing on dolomite rocks and a native and relatively stable karst forest [25]. The Dashahe Nature Reserve has a total area of 270 km$^2$ with a maximum elevation of 1940 m, a minimum elevation of 564 m, and an average elevation of 1252 m. It has a humid monsoon climate with an average annual temperature of 12.1 °C, annual precipitation of 1194 mm, and an annual sunshine duration of 1134 h. The reserve is located in a karst landscape on soluble carbonate rock formations and is extremely rich in biological resources. There are 3594 species of plants in 1082 genera of 296 families and 208 species in 95 genera

of 47 families of macrofungi, making it one of the most valuable gene pools of biological species in the central subtropics of China [26].

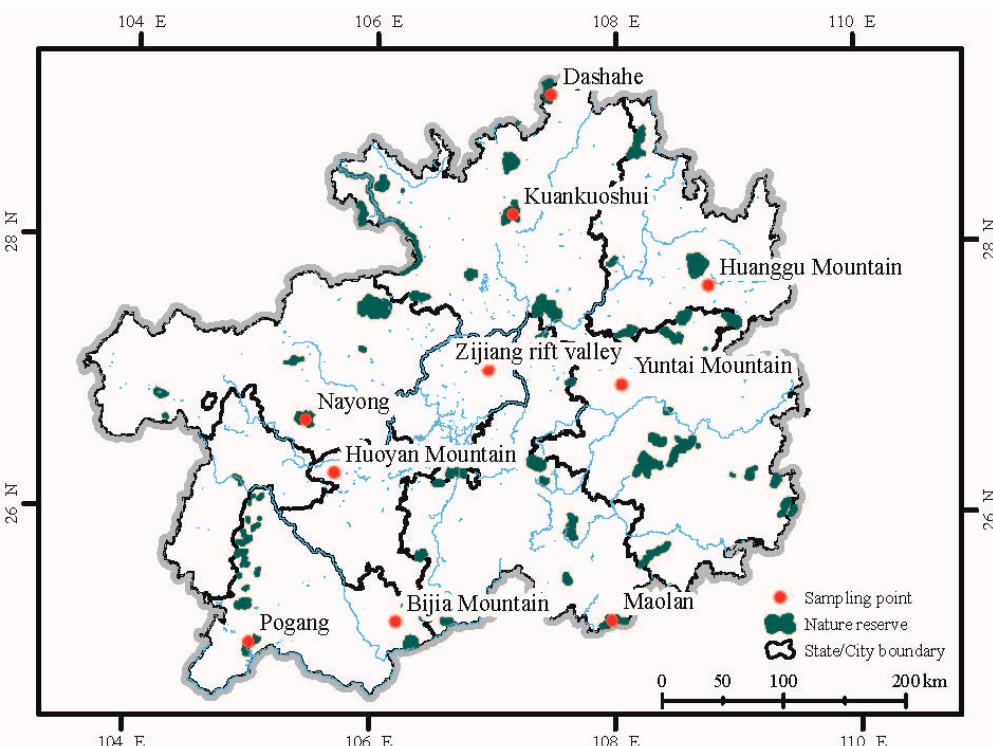

**Figure 1.** Distribution map of sample points.

## 2.2. Research Methodology

### 2.2.1. Sample Site Selection and Vegetation Survey

A total of 57 sample plots were established within seven typical karst forest climax communities, and three study areas in each of four different successional stages were selected from south to north, with three sample plots in each community. The sample plots were 2 m × 5 m for the herb stage, 4 m × 10 m for the shrub stage, 20 m × 20 m for the arbor stage, and 20 m × 20 m for the climax stage. Ten small sample squares were chosen in each sample plot for vegetation surveys. The surveys followed conventional community survey methods [27], where tree plant species, the number of plants, shrub and herb species, and habitat factors including elevation, height, slope degree, slope direction, and soil type were recorded in the sample plots. The information is shown in Tables 1 and 2.

**Table 1.** The basic information of the environmental background of different sample point.

| Area | Succession Stage | Coordinates | Elevation (m) | Precipitation (mm) | Annual Mean Temperature (°C) | Aspect | Dominant Species | Soil Bedrock | Soil Type | Sample Size (m²) |
|---|---|---|---|---|---|---|---|---|---|---|
| Maolan | Herb stage | 108.03 25.26 | 840 | 1590.70 | 19.75 | NW | *Pteridium revolutum, Imperata cylindrical var. major, Pogonatherum crinitum, Trisetum bifidum* | Dolomite limestone | Clay, black limestone soil | 2 × 5 |
| | Shrub stage | 107.94 25.30 | 820 | 1590.70 | 19.75 | SW | *Pyracantha fortuneana, Nandina domestica, Lindera communis, Myrsine semiserrata, Clausena dunniana, Ulmus parvifolia* | Dolomite limestone | Clay, black limestone soil | 4 × 10 |
| | Arbor stage | 107.95 25.29 | 840 | 1590.70 | 19.75 | SW | *Swida wilsoniana, Machilus chienkweiensis, Lindera communis, Cladrastis platycarpa, Choerospondias axillaris* | Dolomite limestone | Clay, black limestone soil | 20 × 20 |
| | Climax stage | 107.99 25.19 | 850 | 1590.70 | 19.75 | SW | *Swida wilsoniana, Pittosporum brevicalyx, Cyclobalanopsis multiervis, Acerwangchii, Carpinus pubescens, Phoebe crassipedicella* | Dolomite limestone | Clay, black limestone soil | 20 × 20 |
| Yuntai Mountain | Herb stage | 108.12 27.18 | 873 | 1083.80 | 17.29 | SW | *Awn, Ophiopogon japonicus, Ficus tikoua Bur., Athyrium dissitifolium* | Carbonate rock | Clay, black limestone soil | 2 × 5 |
| | Shrub stage | 108.16 27.13 | 865 | 1083.80 | 17.29 | NW | *Bridelia tomentosa, Neillia sinensis Oliv., Viburnum dilatatum Thunb., Nothopanax davidii Franch.Harms* | Carbonate rock | Loam, black limestone soil | 4 × 10 |
| | Arbor stage | 108.10 27.10 | 841 | 1083.80 | 17.29 | SW | *Lindera communis Hemsl., Pistacia chinensis Bunge, Quercus acutissima Carr., Platycarya strobilacea* | Carbonate rock | Loam, black limestone soil | 20 × 20 |
| | Climax stage | 108.11 27.12 | 875 | 1083.80 | 17.29 | NW | *Cupressus funebris, Quercus dolicholepis, Platycarya strobilacea, Carpinus pubescens, Quercus phillyraeoides* | Carbonate rock | Loam, black limestone soil | 20 × 20 |

**Table 1.** *Cont.*

| Area | Succession Stage | Coordinates | Elevation (m) | Precipitation (mm) | Annual Mean Temperature (°C) | Aspect | Dominant Species | Soil Bedrock | Soil Type | Sample Size (m²) |
|---|---|---|---|---|---|---|---|---|---|---|
| Dashahe | Herb stage | 107.58 29.15 | 1371 | 1372.20 | 16.81 | NE | *Imperata cylindrica, Carex capilliformis, R. setchuenensis* | Carbonate rock | Clay, black limestone soil | 2 × 5 |
| | Shrub stage | 107.57 29.10 | 1416 | 1372.20 | 16.81 | NE | *Pyracantha fortuneana, Viburnum dilatatum Thunb., R. setchuenensis, Wild persimmon* | Carbonate rock | Clay, black limestone soil | 4 × 10 |
| | Arbor stage | 108.01 29.12 | 1389 | 1372.20 | 16.81 | NE | *Litsea elongata Benth., Machilus versicolora, Carpinus pubescens Burk., Fagus longipetiolata* | Carbonate rock | Loam, black limestone soil | 20 × 20 |
| | Climax stage | 107.58 29.17 | 1304 | 1372.20 | 16.81 | N | *Machilus pingii, Tetracentron sinense, Dipentodon sinicus, Davidia involucrata, Emmenopterys henryi* | Carbonate rock | Loam, black limestone soil | 20 × 20 |
| Nayong | Climax stage | 105.44 26.68 | 1861 | 1226.00 | 14.75 | NW | *Davidia involucrata, Decaisnea insignis, Dipentodon sinicus, Cyclobalanopsis argyrotricha* | Carbonate rock | Loam, black limestone soil | 20 × 20 |
| Pogang | Climax stage | 105.09 25.11 | 1280 | 1501.70 | 17.10 | N | *Eucalyptus robusta, Platycarya strobilacea, Itoa orientalis Hemsl* | Dolomite limestone | Loam, black limestone soil | 20 × 20 |
| Kuankuoshui | Climax stage | 107.06 28.18 | 1450 | 1029.40 | 15.91 | SW | *Fagus longipetiolata, Emmenopterys henryi, Tulip poplar* | Carbonate rock | Loam, black limestone soil | 20 × 20 |
| Huoyan mountain | Climax stage | 105.79 26.47 | 1680 | 1163.10 | 15.98 | W | *Rhododendron stamineum, Birch, Oak* | Carbonate rock | Loam, black limestone soil | 20 × 20 |
| Huanggu mountain | Climax stage | 108.78 27.54 | 1020 | 1542.00 | 17.56 | N | *Fagus longipetiolata, Buxus sinica, Davidia involucrata, Hemlock* | Carbonate rock | Loam, black limestone soil | 20 × 20 |
| Zijiang rift valley | Climax stage | 107.04 26.90 | 720 | 1169.00 | 14.13 | SW | *Betula luminifera, Cinnamomum camphora, Pistacia chinensis, Liquidenbar formosana* | Carbonate rock | Loam, black limestone soil | 20 × 20 |
| Bijia mountain | Climax stage | 106.14 25.12 | 1083 | 1062.70 | 20.53 | NW | *Cyclobalanopsis oak, Carpinus pubescens, Celtis sinensis, Ormosia saxatilis* | Carbonate rock | Loam, black limestone soil | 20 × 20 |

**Table 2.** The basic information of soil physical and chemical properties in different succession stages.

| Area | Succession Stage | pH | BD (g cm$^{-3}$) | SOC (g kg$^{-1}$) | TN (g kg$^{-1}$) | TP (g kg$^{-1}$) | Lci (g C m$^{-2}$) | Ca (g kg$^{-1}$) | Ur (mg g$^{-1}$ 24 h$^{-1}$) | Npa (mg g$^{-1}$ 24 h$^{-1}$) | Sa (mg g$^{-1}$ 24 h$^{-1}$) |
|---|---|---|---|---|---|---|---|---|---|---|---|
| Maolan | Herb stage | 7.34 ± 0.08a | 1.31 ± 0.02a | 28.34 ± 2.80d | 1.57 ± 0.03d | 0.36 ± 0.01d | 12.27 ± 1.08d | 1.38 ± 0.05c | 0.09 ± 0.01c | 0.65 ± 0.08b | 7.08 ± 0.23b |
| | Shrub stage | 7.63 ± 0.07a | 1.25 ± 0.01a | 65.30 ± 4.36c | 6.75 ± 0.43c | 1.01 ± 0.10b | 33.57 ± 2.44c | 3.19 ± 0.48b | 0.89 ± 0.48b | 2.62 ± 0.15a | 7.36 ± 0.14b |
| | Arbor stage | 7.23 ± 0.06a | 1.20 ± 0.01a | 85.22 ± 3.69b | 7.55 ± 0.11b | 0.77 ± 0.02c | 81.76 ± 2.23b | 4.83 ± 0.14a | 1.05 ± 0.11b | 2.16 ± 0.34a | 8.22 ± 0.22b |
| | Climax stage | 7.14 ± 0.27a | 1.02 ± 0.03b | 94.13 ± 3.51a | 8.42 ± 1.10aa | 1.21 ± 0.20a | 141.03 ± 2.53a | 4.83 ± 1.02a | 1.62 ± 0.45a | 2.79 ± 0.33a | 12.77 ± 0.73a |
| Yuntai Mountain | Herb stage | 8.12 ± 0.05a | 1.43 ± 0.06a | 22.52 ± 1.23c | 1.88 ± 0.08d | 0.43 ± 0.01a | 23.33 ± 1.34c | 2.29 ± 0.09d | 0.35 ± 0.04c | 0.62 ± 0.04b | 0.75 ± 0.08c |
| | Shrub stage | 7.94 ± 0.02a | 1.28 ± 0.03b | 41.46 ± 2.05b | 3.67 ± 0.22c | 0.64 ± 0.01a | 40.39 ± 2.19c | 4.49 ± 0.19c | 3.18 ± 0.09b | 2.20 ± 0.07a | 1.36 ± 0.12c |
| | Arbor stage | 7.97 ± 0.04a | 1.19 ± 0.04c | 58.84 ± 3.16b | 5.43 ± 0.25b | 0.57 ± 0.02a | 137.09 ± 6.90b | 6.63 ± 0.14a | 4.18 ± 0.30a | 2.26 ± 0.09a | 9.86 ± 1.99b |
| | Climax stage | 7.93 ± 0.06a | 1.17 ± 0.02c | 82.13 ± 2.48a | 6.75 ± 0.79a | 0.54 ± 0.03a | 181.11 ± 6.72a | 5.09 ± 0.74b | 4.25 ± 0.76a | 2.08 ± 0.24a | 12.83 ± 0.82a |
| Dashahe | Herb stage | 7.49 ± 0.22a | 1.30 ± 0.02a | 18.35 ± 2.13d | 1.84 ± 0.03d | 0.51 ± 0.04a | 21.27 ± 8.71c | 2.26 ± 0.38b | 0.22 ± 0.04c | 0.55 ± 0.08c | 5.87 ± 2.12b |
| | Shrub stage | 6.55 ± 0.04a | 1.18 ± 0.03b | 29.23 ± 3.05c | 2.33 ± 0.17c | 0.37 ± 0.01b | 29.25 ± 2.34c | 1.85 ± 0.13c | 0.35 ± 0.03c | 0.69 ± 0.03c | 4.67 ± 1.51b |
| | Arbor stage | 6.60 ± 0.10a | 1.22 ± 0.02b | 49.93 ± 3.52b | 3.77 ± 0.29b | 0.39 ± 0.01b | 98.86 ± 7.55b | 2.83 ± 0.25b | 0.74 ± 0.10b | 1.89 ± 0.14a | 10.60 ± 1.91a |
| | Climax stage | 7.74 ± 0.09a | 1.12 ± 0.01c | 78.75 ± 2.48a | 5.53 ± 0.67a | 0.69 ± 0.03a | 204.71 ± 12.24a | 3.20 ± 0.31a | 2.28 ± 0.75a | 1.12 ± 0.14b | 11.11 ± 1.74a |

Note: BD is soil bulk density, SOC is soil total organic carbon, TN is total nitrogen, TP is total phosphorus, Lci is litter carbon input, Ca is exchangeable calcium, Ur is soil urease, Npa is neutral phosphatase, Sa is soil sucrase. Different lowercase letters indicate a significant difference ($p < 0.05$) among different succession stages for soil property. Contents are reported as mean ± SE.

### 2.2.2. Soil Sample Collection and Processing

Surface soil samples were collected in each sample plot along two diagonal lines within the square plot using the "S" type five-point mixed sampling method. Three mixed samples were collected from each sample plot for a total of 171 soil samples from 57 sample plots, while 3 soil samples were collected by 100 cm$^3$ foil sampler from each sample plot for a total of 171 and dried for the determination of soil bulk density. Sampling was carried out by removing litter from the surface and removing gravel and roots that were visible to the naked eye after sampling and mixing. One sample from eacg of the Maolan, Yuntai Mountain, and Dashahe Nature Reserves was taken back to the laboratory in a sealed plastic bag, air-dried, and finely ground through a 0.25 mm sieve; the other sample was taken back to the laboratory in a low-temperature sampling box and placed in an ultra-low temperature refrigerator at −70 °C for microbiological determination. Soil samples from the seven climax communities were brought back to the laboratory in sealed plastic bags, air-dried, finely ground, and passed through a 0.25 mm sieve.

### 2.2.3. Litter Sample Collection and Processing

Nylon mesh was used to construct a square sampling frame with an area of 1.0 m$^2$. We randomly selected three small subplots in each plot at the shrub, arbor, and climax stages to sample the litter. The square sampling frame was placed horizontally during sampling. The litter in the square sampling frame was collected every six months. For the herb stage, three small 1.0 m$^2$ subplots were randomly selected in each plot, and the aboveground plant parts were harvested as the litter. A total of 36 samples were collected. All samples were brought back to the laboratory in sealed plastic bags, dried in an oven at 60 °C to a constant weight, and partly ground and passed through a 0.25 mm sieve to determine the carbon content of the litter.

### 2.2.4. SOC Fraction

The wet sieving method of Six et al. [28] was used to determine the SOC fraction. The procedure was as follows: 30 g of air-dried soil sample was passed through a 2 mm sieve and then placed on the top sieve of a microaggregate separator set (top 250 μm sieve, bottom 53 μm sieve). Then, 15 glass beads were added, and after the separator was shaken for 30 min, the >250 μm agglomerates remained on the top sieve; the microaggregate fraction was retained on the 53–250 μm sieve, and soil particles that passed through the 53 μm sieve comprised the fine particle fraction. Then, 25 mL of 0.25 mol/L CaCl$_2$ solution was added to the bucket of the <53 μm sieve, and the mixture was centrifuged at 1730× *g* for 15 min to separate the fine particle fraction. All fractions were transferred to aluminum boxes, steamed using a water bath, and then dried in an oven at 60 °C for 12 h. After drying, the fractions were finely ground and sieved through a 0.25 mm sieve and used to determine the SOC content of each fraction.

### 2.2.5. Methods for Determination of Soil Sample Indicators

Soil physical and chemical properties were determined using the methods described in *Soil Agrochemical Analysis* by Bao Shidan [29]. The specific methods for each soil property were as follows. Soil pH: the potentiometric method with a soil–liquid ratio of 1:2.5; soil bulk density: cutting ring weighing method; SOC: the oil bath heating potassium dichromate oxidative capacity method; soil total nitrogen: the Kjeldahl distillation method; soil total phosphorus: the molybdenum antimony anti-colorimetric method; soil total potassium: the sodium hydroxide fusion-flame photometric method; exchangeable calcium: the ammonium acetate exchange-atomic absorption spectrophotometric method. Soil enzymes were determined by the methods listed in *Soil enzymes and their research methods* by Song-Ying Guan [30]. Soil urease was determined by the phenol-sodium hypochlorite colorimetric method; soil sucrase was determined using the 3,5-dinitrosalicylic acid colorimetric method; soil phosphatase was determined by the sodium phosphate colorimetric method.

### 2.3. Data Processing and Analysis

The SOC maximum saturation capacity model was constructed using the boundary line method of Feng [11] implemented by setting a boundary limit value (top 10%), equivalent to dividing the <53 μm fine particle fraction carbon of the 10 karst forest climax communities into 9 groups and extracting the top 10% of the fine particle fraction. Using the corresponding mass proportions from each group of data, the data were used for linear regression, and the intercept was forced through the zero point to construct a model of the maximum saturation capacity of SOC in karst forests. The maximum saturation capacity of SOC in the three regions was estimated using the regression model. The CSD at each successional stage was defined as the difference between the maximum saturation capacity of SOC and the current SOC.

The data obtained were processed using Excel 2007 software, and statistical analysis was performed using SPSS 19.0 software and Duncan's new complex polar difference method. $p < 0.05$ was considered as significant. Statistical analysis and driver screening were performed using R language software [31].

## 3. Results

### 3.1. Characteristics of Changes in SOC Content

The carbon fractions and SOC contents in different successional stages all showed significant differences (Table 3). The carbon fraction and SOC content of karst forest soil at different successional stages followed the pattern of climax stage > arbor stage > shrub stage > herb stage, and the SOC contents of the soils in the climax stages of Maolan, Yuntai Mountain, and Dashahe were 94.13, 82.13, and 78.75 g C kg$^{-1}$ soil, values that were 3.32, 1.44, and 1.10 times, 3.65, 1.98, and 1.40 times, and 4.29, 2.69, and 1.58 times higher than those in the herb, shrub, and arbor stages, respectively. The SOC content of each successional stage in different karst forests followed the order Maolan > Yuntai Mountain > Dashahe. Among the 10 climax communities, the <53 μm carbon fraction and the SOC content were the highest in Nayong, with 23.56 and 147.11 g C kg$^{-1}$ soil, respectively, and were lowest in Zijiang Geosuture at 8.30 and 42.05 g C kg$^{-1}$ soil, respectively. The proportion of <53 μm organic carbon in SOC did not show significant differences among the 10 climax communities, with a mean value of 19.44%.

**Table 3.** Contents change of soil total organic carbon and fraction carbon.

| Area | Succession Stage | >250 μm Fraction C Content (g C kg$^{-1}$ Soil) | 53–250 μm Fraction C Content (g C kg$^{-1}$ Soil) | <53 μm Fraction C Content (g C kg$^{-1}$ Soil) | Soil Organic C Content (g C kg$^{-1}$ Soil) | Proportion of <53 μm C to Total Organic C (g Fraction 100g$^{-1}$ Soil) |
|---|---|---|---|---|---|---|
| Maolan | Herb stage | 17.33 ± 0.62d | 7.50 ± 0.46d | 3.51 ± 0.38d | 28.34 ± 2.80d | 12.39 ± 0.55c |
| | Shrub stage | 31.91 ± 1.50c | 22.77 ± 1.19c | 10.62 ± 0.68c | 65.30 ± 4.36c | 16.26 ± 0.43b |
| | Arbor stage | 39.49 ± 2.46b | 30.35 ± 2.55b | 15.38 ± 1.29b | 85.22 ± 3.69b | 18.05 ± 0.62a |
| | Climax stage | 43.77 ± 2.35aB | 33.19 ± 2.33aB | 17.17 ± 0.73aB | 94.13 ± 3.51aB | 18.24 ± 1.03aA |
| Yuntai Mountain | Herb stage | 17.32 ± 1.12c | 3.31 ± 0.29d | 1.89 ± 0.19c | 22.52 ± 0.84d | 8.39 ± 0.21b |
| | Shrub stage | 20.70 ± 1.23c | 16.70 ± 1.61c | 4.06 ± 0.21c | 41.46 ± 1.75c | 9.79 ± 0.32b |
| | Arbor stage | 27.58 ± 2.67b | 20.58 ± 3.31b | 10.68 ± 0.98b | 58.84 ± 4.61b | 18.15 ± 0.65a |
| | Climax stage | 38.06 ± 1.48aB | 27.97 ± 1.70aBC | 16.10 ± 2.48aB | 82.13 ± 2.08aC | 19.60 ± 0.24aA |
| Dashahe | Herb stage | 14.69 ± 1.10c | 2.20 ± 0.21d | 1.46 ± 0.19c | 18.35 ± 0.97d | 7.96 ± 0.13c |
| | Shrub stage | 17.46 ± 0.82c | 9.58 ± 0.96c | 2.19 ± 0.24c | 29.23 ± 0.84c | 7.49 ± 0.28c |
| | Arbor stage | 28.93 ± 2.63b | 15.25 ± 1.74b | 5.75 ± 0.52b | 49.93 ± 3.21b | 11.52 ± 0.44b |
| | Climax stage | 39.01 ± 2.01aB | 24.22 ± 2.13aC | 15.52 ± 0.81aB | 78.75 ± 2.45aCD | 19.71 ± 0.36aA |
| Nayong | Climax stage | 69.21 ± 3.42A | 54.34 ± 4.51A | 23.56 ± 3.87A | 147.11 ± 6.55A | 18.42 ± 0.52A |
| Pogang | Climax stage | 41.69 ± 4.74B | 24.65 ± 2.58C | 15.44 ± 1.43B | 81.78 ± 8.89C | 18.88 ± 0.34A |
| Kuankuoshui | Climax stage | 29.06 ± 1.99C | 18.99 ± 1.23D | 11.45 ± 0.28C | 59.50 ± 4.79DE | 19.24 ± 1.17A |
| Huoyan mountain | Climax stage | 28.22 ± 1.53C | 12.18 ± 0.93E | 9.03 ± 1.73D | 49.43 ± 0.63E | 21.09 ± 0.70A |
| Huanggu mountain | Climax stage | 39.65 ± 1.56B | 20.52 ± 0.76D | 15.62 ± 1.77B | 75.79 ± 1.39CD | 20.61 ± 2.26A |
| Zijiang rift valley | Climax stage | 17.95 ± 0.69D | 15.80 ± 0.99DE | 8.30 ± 0.78D | 42.05 ± 1.85F | 19.74 ± 1.76A |
| Bijia mountain | Climax stage | 32.20 ± 2.15C | 20.95 ± 1.64D | 12.26 ± 1.49C | 65.41 ± 6.03D | 18.82 ± 0.25A |

Note: Different lowercase letters indicate a significant difference ($p < 0.05$) among different succession stages, different capital letters indicate a significant difference ($p < 0.05$) among different climax stages for each fraction, C content and proportion are reported as mean ± SE.

### 3.2. CSD Characteristics

The maximum saturation capacity of SOC in karst forests was modelled as $y = 0.66x$ ($R^2 = 0.78$) (Figure 2), and the maximum saturation values of SOC in Maolan, Yuntai Mountain, and Dashahe were 20.70%, 18.88%, and 14.55%, respectively, according to the model. The CSD in karst forests showed significant differences at different successional stages (Figure 3). The CSD in the herb stage was the highest in Maolan, Yuntai Mountain, and Dashahe at 83.04%, 89.99%, and 89.97%, respectively, followed by the shrub stage at 48.69%, 78.50%, and 84.95%, and the lowest in the arbor stage at 25.69%, 43.44%, and 60.49%. The CSD values in the herb stage were 1.71, 1.15, and 1.06 and 3.23, 2.07, and 1.49 times higher than those in the shrub and arbor stages, respectively. The CSD in karst forests showed an overall increasing trend from south to north.

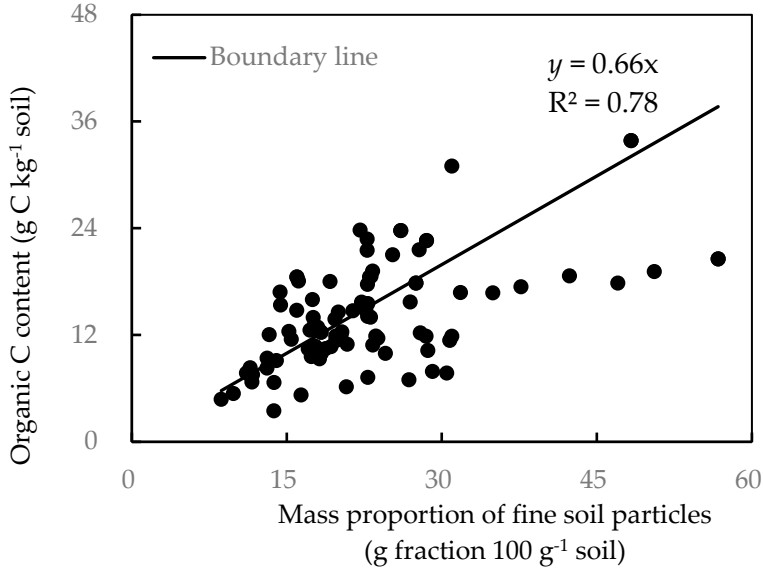

**Figure 2.** Boundary line analysis of organic C content (g C kg$^{-1}$ soil) of <53 μm particles with their mass proportions (g fraction 100 g$^{-1}$ soil) in all bulk soils.

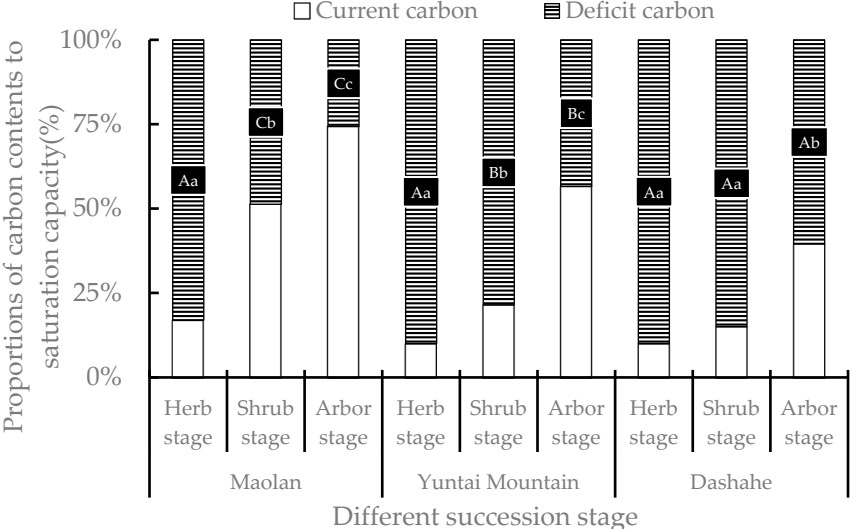

**Figure 3.** Saturated deficit of soil organic carbon at different succession stages in different regions (Note: Different lowercase letters indicate a significant difference ($p < 0.05$) among different succession stages, different capital letters indicate a significant difference($p < 0.05$) among different regions).

### 3.3. Analysis of the Main Drivers of CSD

The main drivers of CSD differed significantly between karst forests (Table 4). In Maolan, the multiple stepwise regression model showed a significant relationship between CSD and litter carbon input, total nitrogen, total phosphorus, and total SOC, with an $R^2$ of 0.96 and a significant model fit ($p < 0.01$). In Yuntai Mountain, the regression model showed that the CSD was linearly related to the amount of litter carbon input ($R^2 = 0.90$; $p < 0.05$). In Dashahe, the stepwise regression model showed a significant relationship between CSD and litter carbon input and neutral phosphatase (corrected $R^2 = 0.91$; $p < 0.05$). All three karst forests showed a significant regression between CSD and litter carbon input, and the larger the coefficient, the smaller the contribution to CSD. The trend of CSD was Maolan > Yuntai Mountain > Dashahe when equal amounts of litter were considered.

**Table 4.** Regression model between soil organic carbon saturation deficit (CSD) and major driving factors.

| Different Regions | Regression Equation | Indicative Factor | *p* Value | Correction $R^2$ |
|---|---|---|---|---|
| Maolan | $y = -0.45x_1 - 6.22x_2 + 43.85x_3 - 0.42x_4 + 94.38$ | $y$: CSD<br>$x_1$: Lci<br>$x_2$: TN<br>$x_3$: TP<br>$x_4$: SOC | 0.001 | 0.96 |
| Yuntai Mountain | $y = -0.40x_1 + 97.15$ | $y$: CSD<br>$x_1$: Lci | 0.000 | 0.90 |
| Dashahe | $y = -0.16x_1 - 11.49x_2 + 98.59$ | $y$: CSD<br>$x_1$: Lci<br>$x_2$: Npa | 0.001 | 0.91 |

Note: CSD is soil organic carbon saturation deficit, Lci is litter carbon input, TN is total nitrogen, TP is total phosphorus, SOC is soil total organic carbon, Npa is neutral phosphatase.

## 4. Discussion

### 4.1. Characteristics of the Changes in SOC Content

The results of this study demonstrated that the carbon fraction and SOC content increased as succession progressed, consistent with the results of other studies [32,33] and indicating that the climax stage is the main carbon sink of karst forest ecosystems and that it has a high carbon sequestration capacity. The main reason for this is that the litter carbon input increased significantly as succession progressed (Table 2), with 12.27 g/m$^2$, 23.33 g/m$^2$, and 21.27 g/m$^2$ in the herb stage of Maolan, Yuntai Mountain, and Dashahe, respectively, reaching 141.03 g/m$^2$, 181.11 g/m$^2$, and 204.71 g/m$^2$ in the climax stage, values that were 11.49 and 7.76 times higher than those in the herb stage. The amount of litter carbon input is the main source of forest soil carbon and directly affects the SOC balance [15,34,35]. Meanwhile, a large number of studies have found that increased carbon input may have a negative excitation effect on soil original organic carbon and inhibit its native soil organic carbon [19,36,37]. The root system is also a source of soil carbon input, and the dominant tree species in both the arbor and climax stages were mixed with evergreen, deciduous broad-leaved trees (Table 1) with well-developed root systems and greater carbon input to the soil. In contrast, the dominant species in the herb and shrub stages were herbs or shrubs (Table 1) with less-developed root systems and less carbon input to the soil. The soil microorganisms play a crucial role in the SOC sequestration process, regulating the SOC balance in both directions [38,39]. The successional sequence from the herb stage to the climax stage can improve the soil agglomeration structure by increasing microbial biomass, especially fungal mycelia; this increases soil organic cementation that in turn increases the physical conservation of SOC [36,40]. The structure and activity of soil microbial communities differ at various successional stages, and fungal cell residues, especially cell wall components, are more difficult to decompose than bacteria. Thus it

is possible that a higher proportion of fungi in the climax stage increases the stability of SOC [6,41].

Many studies have suggested that climatic factors, especially temperature and precipitation, are the most important determinants of SOC distribution at large scales [18,37,42]. The positive relationship between temperature and precipitation and SOC in forests has been highlighted in global and more regional studies [43,44]. In our study, there were significant differences in SOC content among the karst forests with different temperatures and precipitation. Maolan had the highest temperature and precipitation (19.75 °C, 1590.70 mm) (Table 2), and SOC content was also the highest. On the contrary, Dashahe had the lowest temperature (16.81 °C) and average precipitation (1372.20 mm) (Table 2), and SOC content was the lowest. This is consistent with the results from studies on a global scale and numerous regional scales [33,45]. Temperature and precipitation increase could significantly enhance the bio-productivity and accelerate the decomposition rate of litter, thus increasing the input of SOC. When the amount of exogenous carbon input is greater than the amount of SOC mineralization, it is beneficial to the accumulation of SOC [45,46].

### 4.2. Characteristics of the Changes in CSD

CSD is an indicator of the level of future carbon sequestration potential of SOC or the amount of space available for sequestration [47]. The greater the CSD, the greater the potential for future sequestration of SOC. In this study, CSD was highest in the herb stages of Maolan, Yuntai Mountain, and Dashahe at 83.04%, 89.99%, and 89.97%, respectively, followed by 48.69%, 78.50%, and 84.95% in the shrub stage and being lowest in the arbor stage at 25.69%, 43.44%, and 60.49% (Figure 3), indicating that as succession progressed, the CSD decreased. The future carbon sequestration potential gradually decreased, i.e., the herb stage had the most space for carbon sequestration, followed by the shrub and arbor stages. In the three karst forests, the CSD was not significantly different between the herb stages of Maolan, Yuntai Mountain, and Dashahe, while there was a trend of Dashahe > Yuntai Mountain > Maolan at the shrub and arbor stages. On the whole, the CSD showed a gradual increase from south to north, indicating that the future carbon sequestration potential of Dashahe is the largest, followed by Yuntai Mountain and Maolan. On a large spatial scale, Xu et al. [48] found that the CSD in the surface layer of the Daxinganling Forest was 2.20% and 78.80% in the deep layer, indicating that the deep layer had a greater potential for carbon sequestration. Zhang [49] found that the CSD in a protected forest in a desert zone was 27.58%. CSD varies from region to region and is related to the maximum saturation of SOC and the organic carbon content of existing soil mineral particles [9]. In this study, the CSD was related to the organic carbon content of the <53 μm fraction, and the closer the organic carbon content of the <53 μm fraction to the climax stage, the lower the CSD, and the opposite is true. In this study, as succession proceeded, the organic carbon content of the <53 μm fraction increased and was greatest in the karst forest at Maolan, lower at Yuntai Mountain, and least at Dashahe (Table 3). Therefore, regarding the future of karst forests, development should be conducted from north to south as far as possible, and priority should be given to grassland and shrub stages in order to achieve the maximum relative carbon sequestration capacity.

### 4.3. Analysis of the Main Drivers of the CSD

CSD is influenced by multiple interacting factors and involves complex processes [50–53]. In this study, the main drivers of CSD in different karst forests were litter carbon input, total nitrogen, total phosphorus, and total SOC in Maolan, litter carbon input in Yuntai Mountain, and litter carbon input and neutral phosphatase in Dashahe. The results showed that the ecosystem structure of Maolan was the most complex among the different karst forests, and there was an overlap of environmental factors. Therefore, the CSD was influenced by multiple factors. Yuntai Mountain and Dashahe have relatively simple ecosystem structures, so there are few controlling factors. Litter carbon input is the main driving factor of CSD in karst forests, reflecting the important role of litter in maintaining soil carbon

balance in karst forests. As the main source of the soil carbon pool, litter decomposition and accumulation will affect the dynamic balance of SOC [54,55]. The amount of carbon input to litter is controlled by litter quality and external environmental conditions, and these two factors can be adjusted to control the litter carbon input [56,57]. Litter regulates CSD by affecting multiple factors such as soil microbial community structure and native soil organic carbon excitation. Phosphatase is one of the most active enzymes in soil. It is an important indicator enzyme for the characterization of soil biological activity, and it plays an important role in soil phosphorus cycling [58]. In this study, neutral phosphatase might affect carbon sequestration by changing soil fertility or might affect the amount of litter carbon input by changing the phosphorus absorption capacity of plants, thus affecting CSD. The results of this study were generally consistent with those of other researchers. Tian et al. [59] found that the factors affecting SOC stability at different elevation gradients were temperature, litter, and soil physicochemical properties. Liu et al. [60] found that the main controlling factors of SOC in Moso bamboo forests were soil porosity, capacitance, and soil enzyme activity. Guan et al. [45] found that the main influencing factors of SOC in northwestern forest ecosystems were standing age, temperature, humidity, elevation, and litter. In conclusion, although the main drivers of CSD in karst forests vary, the core driver is the amount of litter carbon input; therefore, this factor could be adjusted to regulate CSD in the future.

## 5. Conclusions

By constructing the maximum saturated capacity model of SOC in karst forests, we estimated the saturated deficit of SOC in different regions and succession stages and analyzed its main driving factors. The soil carbon fraction and SOC content in karst forests followed the pattern climax stage > arbor stage > shrub stage > herb stage, and the SOC content in different karst forests was Maolan > Yuntai Mountain > Dashahe. The CSD in the herb stages of Maolan, Yuntai Mountain, and Dashahe were the highest, and the future carbon sequestration potential of the herb stage was increased from south to north, with greater potential for exploitation. The core driver of CSD in forest ecosystems of the karst forests is the amount of litter carbon input, which can be adjusted to control CSD in karst forests.

**Author Contributions:** Conceptualization, L.Z. and L.Y.; methodology, J.C. and Y.W.; formal analysis, L.Z.; investigation, F.L. and L.F.; resources, L.Y.; writing—original draft preparation, L.Z.; writing—review and editing, L.Z.; supervision, L.Y.; funding acquisition, L.Y. All authors have read and agreed to the published version of the manuscript.

**Funding:** This research was funded by the Project of National Key Research and Development Program of China (grant number 2016YFC0502604); the Application Foundation Major Project of Guizhou Province of China (Qian Ke He JZ [2014] 2002); the Construction Program of Biology First-class Discipline in Guizhou of China (GNYL [2017]009); and the Project of Promoted Innovation of Colleges and Universities of Guizhou Province of China (Qian Jiao He Collaborative Innovation [2014]01).

**Institutional Review Board Statement:** No animal interventions were used in this research.

**Data Availability Statement:** All data are available on request.

**Acknowledgments:** We sincerely thank Meng Xu and his team for their help and support during the manuscript writing process.

**Conflicts of Interest:** The authors declare no conflict of interest.

## Abbreviations

SOC: Soil organic carbon; CSD: Carbon saturation deficit.

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
