# Peer review of "Characteristics and Drivers of Soil Organic Carbon Saturation Deficit in Karst Forests of China"

_diversity, doi:10.3390/d14020062_

Round 1
Reviewer 1 Report
The article “Characteristics and Drivers of Soil Organic Carbon Saturation Deficit in Karst Forests of China” has been drafted nicely and provides a new set of information in the soil carbon research.
I have few suggestions that may be considered during the revision process
- The descriptions of the Karst soil provided in the Introduction and methodology section is adequate. However, this can be further enhanced by incorporation soil type (as per IUSS) and dominant soil texture classes. Authors may incorporate information on this in the revised manuscript.
- In the Introduction section, IPCC default SOC saturation values may be incorporated to enrich information on carbon saturation deficit.
- In sub-section 2.2.4., clearly mention whether the dry or the wet sieving methods were used for aggregate separation.
- The soil sampling strategy for bulk density is missing in the manuscript. Similarly, the methods used for estimation bulk density is also missing. Please provide information on soil sampling and methodology used for bulk density.
- Provide information on methodology used for the TOC and SOC estimation.
- The study area map is not clear. Please consider replacing it with a better version.
- Discussion section can be further improved by incorporating case studies from other parts of the world
- Conclusion section is reading like more of a result section. This should highlight the salient findings of the study. Also include the limitation and future prospect of the study in this section.
Author Response
Dear reviewer,
Thank you for taking time out of your busy schedule to review my manuscript so soon, I will be very grateful. I have modified the manuscript according to your opinions. Please see the attachment.
Best regards,
Limin Zhang
January 6, 2022

Reviewer 2 Report
Dear authors,
The introduction could be improved by better explaining the meaning of "Soil Organic Carbon Saturation" and "Soil organic carbon saturation deficit (CSD)".
There are some corrections proposed.
Line 19: “An SOC” or “A SOC”.
Lines 111-117: It is written “height under branches, plant height, diameter at breast height (for trees above 3 cm), crown width, …The information is shown in Tables 1 and 2”. In Table 1 we can find the dominat species and in Table 2 soil properties. However, height under branches, plant height, diameter at breast height (for trees above 3 cm) and crown width are not in Tables 1 or 2.
Line 123: soil capacity measurement? What do you mean by “soil capacity”? Do you mean “soil properties”?
Line 145: I think the right term is “microaggregate” instead of “microagglomerate”. Please review the manuscript in this sense.
Line 157: “factor” or “soil property”?
Line 159-159: It is not clear to me how the authors measure the SOC (analytical method). They mention in the manuscript TOC and SOC. In Material and Methods, they introduce: “soil organic matter: the oil bath heating potassium dichromate oxidative capacity method”. I suppose the authors want to mean that TOC and SOC were analyzed with the “oil bath heating potassium dichromate oxidative capacity method”. Please, clarify.
Line 197 (Table 1): The scientific names of species should be italicized.
Lines 203-204: In the note of the Table appears “BS is bacterial Shannon 203 index, FS is fungi Shannon index”. However, these indexes are not in the Table. In this sense, perhaps the sentence “The soil microorganisms were determined by bacterial and fungal community structure in combination with high throughput sequencing. The samples were sent to Shanghai Meiji Biomedical Technology Co., Ltd. for testing” should be removed from Material and Methods (Lines 163-165).
“mg g-1 24h-1” instead of “mg g-1 24h”
Line 225: “Different lowercase letters indicate a significant difference (P<0.05) among different succession stages for each fraction” or something like that instead of “Different lowercase letters indicate a significant difference (P<0.05) among different succession stages”.
Line 238 (Figure 2): It is not necessary to have two decimals on the axes when the decimals are zero.
Line 261: Please, add a note to the Figure with the abbreviations.
Line 265: The word “components” is not right here. When we talk about soil components we refer to organic matter, water, air, …
Line 275: What does it mean “soil proto-organic carbon”? I could not read references 19 and 37 because of the language but I could not find this term in reference 36.
Line 322: “in a protected forest” instead of “in the protected forest”.
Lines 351-353: How do you know that neutral phosphatase has changed soil fertility and has altered plant phosphorus uptake capacity?.
I think it is very risky to affirm “In this study, neutral phosphatase affected CSD by altering soil phosphorus levels, on the one hand by changing soil fertility to affect SOC sequestration and on the other hand by altering plant phosphorus uptake capacity to affect litter carbon input to affect CSD”
Perhaps you can use the verbs “could” “might”, …
Author Response

(The authors gave the same response as above.)

Reviewer 3 Report
Review for “diversity-1545710”
Characteristics and Drivers of Soil Organic Carbon Saturation Deficit in Karst Forests of China
This manuscript dealt with an interesting topic about soc and its sequestration process in Karst Forests. The manuscript, except in the introduction section well designed and the methodology well designed. However, there are still a few comments which need to be addressed by the authors before publication. Please see below for detailed comments.
L18-20: the method should be clearly explained, such as the number of samples, depth of sapling, method of sampling collection, lab analysis.
L33-45: soil organic carbon is important to soil health and quality indicators and in all types of soil plays the main role, so I suggest not limiting the importance of SOC in a specific land cover, for clarification; see the following example, and it is worth to include some recent studies as well
https://doi.org/10.1016/j.still.2012.01.011
https://doi.org/10.1016/j.geodrs.2020.e00256
https://doi.org/10.1016/j.catena.2021.105723
the introduction section needs to be restructured, and the current format contains one paragraph for the whole topic. I suggest dividing it into 3-5 paragraphs and developing them.
Please move figure 1 just after L 103.
Author Response

(The authors gave the same response as above.)

Round 2
Reviewer 2 Report
The manuscript has been improved according to the suggestions of the reviewer. They authors could have improved the introduction by adding more information related with soil organic carbon saturation and more references to current articles as they were given the opportunity.